# Virtual Screening, Structural Analysis, and Formation Thermodynamics of Carbamazepine Cocrystals

**DOI:** 10.3390/pharmaceutics15030836

**Published:** 2023-03-03

**Authors:** Artem O. Surov, Anna G. Ramazanova, Alexander P. Voronin, Ksenia V. Drozd, Andrei V. Churakov, German L. Perlovich

**Affiliations:** 1G.A. Krestov Institute of Solution Chemistry RAS, 153045 Ivanovo, Russia; 2Institute of General and Inorganic Chemistry RAS, Leninsky Prosp. 31, 119991 Moscow, Russia

**Keywords:** cocrystal, carbamazepine, screening, thermodynamics, solubility

## Abstract

In this study, the existing set of carbamazepine (CBZ) cocrystals was extended through the successful combination of the drug with the positional isomers of acetamidobenzoic acid. The structural and energetic features of the CBZ cocrystals with 3- and 4-acetamidobenzoic acids were elucidated via single-crystal X-ray diffraction followed by QTAIMC analysis. The ability of three fundamentally different virtual screening methods to predict the correct cocrystallization outcome for CBZ was assessed based on the new experimental results obtained in this study and data available in the literature. It was found that the hydrogen bond propensity model performed the worst in distinguishing positive and negative results of CBZ cocrystallization experiments with 87 coformers, attaining an accuracy value lower than random guessing. The method that utilizes molecular electrostatic potential maps and the machine learning approach named CCGNet exhibited comparable results in terms of prediction metrics, albeit the latter resulted in superior specificity and overall accuracy while requiring no time-consuming DFT computations. In addition, formation thermodynamic parameters for the newly obtained CBZ cocrystals with 3- and 4-acetamidobenzoic acids were evaluated using temperature dependences of the cocrystallization Gibbs energy. The cocrystallization reactions between CBZ and the selected coformers were found to be enthalpy-driven, with entropy terms being statistically different from zero. The observed difference in dissolution behavior of the cocrystals in aqueous media was thought to be caused by variations in their thermodynamic stability.

## 1. Introduction

The development of multicomponent crystals has been recognized as one of the most versatile and robust routes to modify the functional properties of solid-state materials via the deliberate manipulation of intermolecular interactions and the alteration of the crystalline environment through the selection of suitable coformers [1,2]. In particular, in the field of pharmaceutics, the cocrystallization approach can offer a significant improvement in a range of physicochemical properties of an active pharmaceutical ingredient (API), leading to a novel formulation with enhanced therapeutic efficacy [3,4,5,6,7,8]. However, because a variety of pharmaceutically acceptable coformers are often accessible for cocrystallization with a certain API, the problem of rational design of pharmaceutical cocrystals is highly relevant. Experimental screening of cocrystals is usually associated with a time-consuming trial-and-error procedure that requires several experimental techniques and involves a limited range of available coformers [9]. Therefore, during the last two decades, different qualitative and quantitative theoretical strategies enabling rational pre-selection of the most promising coformers before real experimental screening have been developed and explored [10]. One of the most commonly utilized methods for selecting coformers for cocrystallization is the supramolecular synthon approach [11,12]. This technique, however, is difficult to apply for forecasting new multicomponent systems with no pronounced specific interactions [13] or competition between them [14] due to its descriptive nature and lack of quantitative criteria. Indeed, Cappuccino et al. recently demonstrated that predictions based on known supramolecular synthons and their relative frequencies taken from the CSD are no better than a random screening search [15]. Hence, a number of more reliable quantitative computational tools for coformer selection have been presented. These may be generally classified [16] as knowledge-based (both structural informatics [17,18] and thermodynamic methods [19,20]), physics-based (Hansen solubility parameters [21], molecular electrostatic potential map [22,23], COSMO-RS [24], and crystal structure prediction [25,26,27]), and machine learning methods [28,29,30,31,32]. Each of the listed techniques has its own merits and drawbacks, while the success rate of prediction results may vary significantly, depending on an API and the size of a test set [33].

For example, the predictive strength of several virtual screening methods has been tested on the cocrystals of the well-known anticonvulsant drug carbamazepine (CBZ) (Figure 1). Salem et al. [34] examined the efficacy of Hansen solubility parameters as descriptors for a cocrystal screening of CBZ using a set of 60 positive and negative coformers, with an accuracy of around 80%. The COSMO-RS theory has been utilized to perform affinity predictions for CBZ combinations with 75 coformers, demonstrating that the proportion of known CBZ cocrystals generally declines with increasing values of the calculated excess enthalpy parameter [35]. Ahmadi et al. reported that the molecular electrostatic potential surface (MEP) approach predicted 87% of the CBZ cocrystals using a set of 29 known binary systems, though no negative cocrystallization results were considered in this study [33]. Crystal structure prediction methods were also used to explain the difference in outcomes of CBZ cocrystallization with isonicotinamide and picolinamide [36], as well as to estimate the possibility of the formation of CBZ cocrystals with ten preselected candidate coformers [27]. Devogelaer et al. [29] created a machine learning method that achieved an accuracy of 80% and a precision value of 79% for a set of 50 experimentally tested combinations of CBZ with various coformers. Jiang et al. [32] introduced an alternative machine learning approach based on graph neural networks that exhibited 100% predictive performance in terms of the balanced accuracy metrics on a relatively small subset of CBZ combinations with 22 coformers.

In this study, we used a library of 17 coformers that have never been tested against CBZ in order to extend the current list of positive and negative outcomes of the cocrystallization experiments for the drug. As a result, three new CBZ cocrystals were obtained, and two of them were structurally characterized. We also applied three virtual screening models belonging to fundamentally different classes (i.e., hydrogen bond propensity [18], molecular electrostatic potential map [22], and machine learning method CCGNet [32]) and compared their performance metrics for the experimentally verified list of the CBZ cocrystallization results with 87 distinct compounds.

Another aspect the current study aims to investigate is the thermodynamic stability of the CBZ cocrystals relative to their corresponding precursors, which is typically quantified in terms of the Gibbs energy change of the cocrystal formation reaction (Δ_form_G) [37,38,39,40]. Unfortunately, the thermodynamic parameters of the cocrystallization process are rarely discussed in the literature. Currently, only a few experimental investigations that deal with the thermodynamics of a multicomponent crystal formation have been comprehensively addressed utilizing Gibbs energy and both its enthalpic and entropic contributions [40,41,42,43,44,45,46], as opposed to the numerous structural studies of cocrystals. Although theoretical works assert that the enthalpic term controls the stability of multicomponent crystals [47,48], experimental evidence suggests that the relative contributions of the enthalpic and entropic components to the driving force can vary significantly, resulting, in some cases, in the formation of entropically favorable cocrystals [43]. In addition, knowledge of the enthalpy and entropy of the cocrystallization reaction is crucial for predicting how the relative thermodynamic stability of the cocrystal changes with temperature. This work is a continuation of our systematic investigation into the thermodynamic aspects of the formation of the CBZ cocrystals [46]. Herein, we focus on the new cocrystals of the drug with the positional isomers of acetamidobenzoic acid (Figure 1), which were identified by unbiased experimental screening and thoroughly investigated in terms of structural characteristics, formation thermodynamics, and aqueous solution stability.

According to a survey from the Cambridge Structural Database (CSD), the cocrystal formation between CBZ and compounds bearing the acetamido group has not been thoroughly investigated, as no instances of such multicomponent crystals can be found. The single reported attempt to combine carbamazepine with acetaminophen (4-acetamidophenol) failed to produce a new solid phase and resulted in a simple physical mixture [35]. Therefore, investigation of new multicomponent crystals of CBZ with isomers of acetamidobenzoic acids is of great importance in terms of the fundamental understanding of the cocrystallization process, as the identification of the key supramolecular synthons responsible for the stabilization of the crystalline environment is the basis of crystal engineering [12,49]. Further exploration of the rich structural landscape of this drug may help to find and rationalize important structure-property relationships between the packing features, strength and patterns of intermolecular interactions in solid forms, and various pharmaceutically relevant physicochemical parameters. Although acetamidobenzoic acids are rarely used as coformers in the cocrystallization trials, 4-acetamidobenzoic acid has been employed to obtain new pharmaceutical cocrystals with different drugs, such as linogliride [50], nicotine [51], and upadacitinib [52]. In addition, in the human body, 4-acetamidobenzoic acid (Acedoben) is recognized as a metabolite of such drugs as 4-aminobenzoic acid [53] and benzocaine [54]. The compound is also a component of several pharmaceutical formulations, including isoprinosine [55,56] and deanol acetamidobenzoate [57].

## 2. Materials and Methods

### 2.1. Compounds and Solvents

Carbamazepine (C_15_H_12_N_2_O, 98%) was purchased from Acros Organics (Pittsburgh, PA, USA) and identified as a polymorphic form III based on its single crystal structure data (CBMZPN01). All the coformers were purchased from Sigma-Aldrich (St. Louis, MO, USA) or Acros Organics (Pittsburgh, PA, USA). The materials were used as received. The solvents were of analytical or chromatographic grade.

### 2.2. Cocrystal Synthesis

The grinding experiments were performed using a planetary micro mill, Pulverisette 7 (Fritsch, Idar-Oberstein, Germany), in 12 mL agate grinding jars with ten 5 mm agate balls at a rate of 500 rpm for 60 min. In a typical experiment, 80–100 mg of the physical mixture of CBZ and a coformer in a 1:1 molar ratio was placed into a grinding jar, and 50–60 μL of an organic solvent (acetonitrile, methanol, and ethanol) were added with a micropipette. The bulk samples of the CBZ cocrystals with 3- and 4-acetamidobenzoic acids were also prepared by the slurry method. In brief, 100 mg of CBZ and an equimolar amount of the acid were stirred in methanol or acetonitrile for 24 h. Bulk samples were filtered and dried at room temperature for 12 h. The phase purity of the resulting powder samples was confirmed by powder X-ray diffraction by comparing the experimental powder patterns with those calculated from the crystal structures (Appendix A).

### 2.3. Solution Crystallization

The single crystals of the new CBZ cocrystals were successfully obtained by the slow evaporation method. The [CBZ + 3AcAmBA] (1:1) cocrystal was prepared by dissolving a physical mixture in a 1:2 molar ratio in acetone, while the [CBZ + 4AcAmBA] (1:1) cocrystal was prepared by dissolving a physical mixture in a 1:1 molar ratio in ethyl acetate. The resulting solutions were filtered into 10-mL vials, covered by a perforated parafilm with a few small holes, and allowed to evaporate slowly at room temperature (*ca.* 23 °C) until a crystalline material was formed.

### 2.4. Single Crystal and Powder X-Ray Diffraction

The single crystal diffraction data for the cocrystals were collected using a Bruker SMART APEX II diffractometer (Bruker AXS, Karlsruhe, Germany) with graphite-monochromated Mo-K*α* radiation (*λ* = 0.71073 Å). Absorption corrections based on measurements of equivalent reflections were applied [58]. The structures were solved by direct methods and refined by full matrix least-squares on *F*^2^ with anisotropic thermal parameters for all the non-hydrogen atoms [59]. All the hydrogen atoms were found using the Fourier difference maps and refined isotropically. In the structure of 2AcAmBA ethyl acetate solvate, the electron density of highly disordered EtOAc molecules was removed by the SQUEEZE procedure (113 electrons per unit cell) [60]. The crystallographic data were deposited with the Cambridge Crystallographic Data Centre as supplementary publications under the CCDC numbers 2241313, 2241312 and 2241311 for [CBZ + 3AcAmBA] (1:1), [CBZ + 4AcAmBA] (1:1), and 2AcAmBA ethyl acetate solvate, respectively. This information can be obtained free of charge from the Cambridge Crystallographic Data Centre at www.ccdc.cam.ac.uk/data_request/cif (accessed on 11 February 2023).

The laboratory PXRD data of the bulk materials were recorded under ambient conditions on a D2 Phaser (Bragg-Brentano) diffractometer (Bruker AXS, Karlsruhe, Germany) with a copper X-ray source (λ_CuKα1_ = 1.5406 Å) and a LYNXEYE XE-T high-resolution position-sensitive detector. The samples were placed into the plate sample holders and rotated at a rate of 15 rpm during the data acquisition.

### 2.5. Thermal Analysis

The thermal analysis was carried out using a differential scanning calorimeter with a refrigerated cooling system (Perkin Elmer DSC 4000, Waltham, MA, USA). The sample was heated in a sealed aluminum sample holder at a rate of 10 °C·min^−1^ in a nitrogen atmosphere. The unit was calibrated with indium and zinc standards. The accuracy of the weighing procedure was ± 0.01 mg.

TG experiments were performed using a TG 209 F1 Iris thermomicrobalance (Netzsch, Selb, Germany). Approximately 10 mg of the bulk sample was added to a platinum crucible. The samples were heated at a constant rate of 10 °C·min^−1^ and purged throughout the experiment under a dry argon stream at 30 mL·min^−1^.

### 2.6. Solubility Experiments

The saturation shake-flask method was used to measure the aqueous solubility of the CBZ cocrystals in a pH 2.0 buffer solution at the eutectic point, where drug and cocrystal are in equilibrium with the solution. The eutectic point between the cocrystal and the drug was reached by suspending 80 mg of the cocrystal and 30 mg of CBZ in 2 mL of the buffer solution. The obtained suspension was stirred continuously for 72 h at 37.0 ± 0.1 °C. After equilibration, the final pH of the solution was measured. The suspension was diluted with the mobile phase after being filtered through a 0.22-m PTEF filter. Concentrations of compounds were analyzed by HPLC. The solid phases after the experiment were characterized by PXRD.

The solubility of the cocrystals and their constituents was also measured in organic solvents at 20.0, 25.0, 30.0, 35.0, and 40.0 ± 0.1 °C. For the [CBZ + 3AcAmBA] (1:1) cocrystal, methanol was used. For [CBZ + 4AcAmBA] (1:1), all the experiments were performed in acetonitrile. An excess of the solid was placed in an Eppendorf tube, and 2 mL of solvent was added. The resulting suspension was shaken in a ThermoMixer C (Eppendorf, Hamburg, Germany) at 800 rpm for 48 h at the specified temperature. Subsequently, the saturated solution was separated using a 0.22 μm PTEF filter, diluted, and analyzed using HPLC. The solid phases after the experiment were collected and characterized by PXRD. Each experiment was repeated three times.

### 2.7. High-Performance Liquid Chromatography (HPLC)

The HPLC was performed on an LC-20 AD Shimadzu Prominence model (Shimadzu, Kyoto, Japan) equipped with a PDA detector and a Luna C-18 column (150 mm × 4.6 mm i.d., 5 μm particle size, and 100 Å pore size). The column temperature was set to 40 °C. To achieve the proper separation of the cocrystal components, different ratios of the mobile phase, consisting of acetonitrile and a 0.1% aqueous solution of trifluoroacetic acid (35:65 *v*/*v*), were used. The flow rate was 1.0 mL·min^−1^, and the injection volume was 20 μL. The UV detection of carbamazepine, 2-acetamidobenzoic acid, 3-acetamidobenzoic acid, and 4-acetamidobenzoic acid was carried out at the wavelengths of 285 nm, 249 nm, 220 nm, and 266 nm, respectively.

### 2.8. Virtual Cocrystal Screening Methods

#### 2.8.1. Hydrogen Bond Propensity (HBP)

The hydrogen bond propensity calculations [18,61] were performed using a Python script available on the official CSD GitHub repository (https://github.com/ccdc-opensource, accessed on 2 March 2023) and the CSD Python API. All the calculations were performed with the latest version of the CSD database (ver. 5.43) [62] using a predefined library of coformers. The calculated values of the multicomponent score, i.e., the difference between the propensity of the best hetero-interaction and the best homo-interaction, were applied to estimate the probability of cocrystal formation [63,64]. The multicomponent score value ≥ 0 indicates that the hetero-interaction formation prevails and a cocrystal is likely to be formed. In the case where the multicomponent score ≤ 0, homo-interactions prevail and, therefore, cocrystallization is unlikely to occur [64].

#### 2.8.2. Co-Crystal Graph Network (CCGNet)

CCGNet is a deep learning framework for virtual screening of binary organic cocrystals, which has recently been published by Jiang et al. [32]. The model was trained using a large cocrystal dataset of 7871 samples (6819 positive samples and 1052 negative samples), with the molecular graph representation of substances and 12 molecular descriptors related to the cocrystal formation serving as complementing features. In contrast to previously reported machine learning models proposed to predict cocrystallization outcome [29,30], Jiang et al. made the published version of the CCGNet model freely available via a specially compiled Python script, allowing anyone to replicate the prediction results shown in the original paper as well as to perform virtual screening for the pre-defined compound pairs. In this work, the performance of the CCGNet model was tested on the experimentally verified set of CBZ-coformer combinations. 3D structures of coformers for both positive and negative samples were taken from PubChem and stored as *.sdf files. No geometry optimizations of the coformers molecular structures were performed to simulate the high-throughput screening conditions.

#### 2.8.3. Molecular Electrostatic Potential Surface (MEP)

The molecular electrostatic potential surfaces [65] of the CBZ and coformers used in the virtual screening were generated at the B3LYP/def2-tzvp level of theory using Gaussian09 [66]. All the calculations were performed using geometry-optimized structures. The local maxima and minima sites on the molecular electrostatic potential surfaces were extracted using MultiWFN (ver. 1.8) [67]. The values of the local maxima and minima were converted into the corresponding hydrogen bond donor and acceptor interaction site parameters (α and β) according to the equations provided by Hunter et al. [22] These parameters were further used in the computation of site pair interaction energies (E) for homomeric and heteromeric molecular pairs as described elsewhere [22,68]. All the calculations were performed assuming the 1:1 molar ratio of the components in a cocrystal. The probability of cocrystal formation was estimated based on ΔE values:ΔE = E_CC_ − E_1_ − E_2_,(1)
where E_1_, E_2_, and E_CC_ are the interaction site pairing energies of the pure solids (solid 1 and solid 2) and the cocrystal, respectively. ΔE < 0 indicates a high probability of cocrystal formation. In the case of ΔE > 0, cocrystallization is unlikely to occur. The MEP surface map and selected extrema were plotted on an electron density ρ = 0.002 isosurface, visualized and rendered using the VMD program (ver. 1.9.3) [69], and post-processed according to a tutorial provided by Tian Lu [67].

### 2.9. Computational Methods

#### 2.9.1. Periodic DFT Calculations

The periodic DFT computations with localized Gaussian basis sets were performed using the CRYSTAL17 software v.1.0.2 [70] at the B3LYP-D3(BJ,ABC)/6-31G(d,p) level of theory [71,72,73]. It was demonstrated that this level of theory provided reliable and consistent results when studying the non-covalent interactions in organic crystals [74]. The unit cell parameters of the crystals obtained in the X-ray diffraction experiment were fixed, and the structural relaxations were limited to the positional parameters of the atoms. An experimental crystal structure with normalized X–H bond lengths was used as the starting point for geometry optimization. The shrinking factor reflecting the density of the k-point grid in reciprocal space was set at least to 4, depending on the reciprocal lattice vectors in a particular crystal. The TOLDEE and TOLINTEG parameters were set to 10 and 7 7 7 7 25, respectively. All the normal vibrational modes for the relaxed structures were found to have positive frequencies, which is an indicator of the local minimum on the potential energy surface. Further computational details have been described in our previous works [75,76].

#### 2.9.2. Noncovalent Interaction Energies and Lattice Energy Calculation

In order to quantify the energies of particular noncovalent interactions in the crystal, a Bader analysis of the periodic electron density in the crystal (or QTAIMC) was performed in the TOPOND software v.1.0.2 [77] currently built into the CRYSTAL suite. The search for (3, −1) critical points was conducted between the pairs of atoms within the 5 Å radius, and the interactions with positive Laplacian and electron density, ρ_b_, at the (3, −1) point higher than 0.003 a.u. were taken into consideration.

The energy of a particular noncovalent interaction, E_int_, was evaluated based on the local kinetic energy density at the (3, −1) critical point (G_b_) by the correlation equation proposed in the work by Mata et al. [78]:E_int_ (kJ·mol^−1^) = 1147 · G_b_ (a.u.)(2)

Equation (2) yields reasonable E_int_ values for molecular crystals with different types of intermolecular interactions, including conventional and non-conventional hydrogen bonds, halogen bonds, etc. [79,80,81].

## 3. Results and Discussion

### 3.1. Experimental Screening via the Mechanochemical Method

A survey of the most recent version of CSD [62] (CSD version 5.43) has revealed that there are more than 60 distinct multicomponent crystalline forms of CBZ, including cocrystals, cocrystal polymorphs, cocrystal solvates/hydrates, as well as cocrystals with multiple stoichiometries (excluding solvates or hydrates of CBZ itself). To further extend the range of CBZ solid forms, we constructed a custom library of 17 coformers that contained the same combinations of functional groups as in the compounds that have already been known to cocrystallize with carbamazepine. The majority of the selected coformers are aromatic carboxylic acids, with a few benzamide derivatives included to diversify the set (the full list of the coformers used and the results of the experimental cocrystal screening are provided in Appendix A). It is important to emphasize that no virtual screening approaches were applied prior to experimental screening to minimize bias towards the coformer selection. All cocrystallization experiments were carried out using a physical mixture of components in a 1:1 molar ratio via liquid assisted grinding in the presence of methanol, ethanol, or acetonitrile. The PXRD and DSC techniques were used to identify the resultant products and evaluate their phase purity. According to the received screening results, only three compounds out of 17 were found to produce novel multicomponent forms with CBZ, specifically three isomers of acetamidobenzoic acid (2AcAmBA, 3AcAmBA, and 4AcAmBA). PXRD patterns for the products of mechanochemical treatment of CBZ and acetamidobenzoic acids demonstrated the formation of new crystalline phases distinct from the starting components (Appendix A). The formation and thermal behavior of the [CBZ + 2AcAmBA] (1:1), [CBZ + 3AcAmBA] (1:1), and [CBZ + 4AcAmBA] (1:1) cocrystals were further supported by the DSC and TG measurements (Appendix A). The results of the thermal analysis indicated that cocrystallization of CBZ with the selected acetamidobenzoic acids led to the formation of non-hydrated/non-solvated crystalline products, as the DSC thermograms for the cocrystals contained only one sharp endotherm different from their starting materials and no other phase transitions were observed. The TGA analysis did not show a weight loss before the melting of the cocrystals. According to the onset melting temperatures, [CBZ + 2AcAmBA] (1:1) and [CBZ + 3AcAmBA] (1:1) showed lower melting points relative to those of pure constituents, while [CBZ + 4AcAmBA] (1:1) melted at a slightly higher temperature than CBZ (Appendix A). Beyond the melting temperatures, a rapid weight loss step was observed, which was attributed to the degradation of the CBZ cocrystals (Appendix A).

### 3.2. Virtual Screening of CBZ Cocrystals

As mentioned above, carbamazepine is one of the most extensively studied drug compounds in the realm of pharmaceutical cocrystallization. The CSD, which currently contains more than 60 instances of CBZ cocrystals, cocrystal polymorphs, and cocrystal solvates, is the primary source of successful cocrystallization results [62]. However, when evaluating the prediction power of theoretical (or virtual) cocrystal screening tools, a trustworthy collection of negative combinations of an API and coformers is equally important. In this work, negative samples of CBZ-coformer pairings were acquired from the literature and supplemented with our findings. Taking together both favorable (55) and negative (32) instances, 87 cases of cocrystallization outcomes, were considered (Appendix A). Three distinct methods of virtual screening were applied and compared for accuracy in forecasting CBZ cocrystals. The following techniques were used: the hydrogen bond propensity approach (HBP) [18,61], the method based on the analysis of molecular electrostatic potential surface values (MEP) [22], and the recently reported machine learning approach that takes advantage of the combined usage of the graph neural network framework and molecular descriptors to predict the formation of cocrystals (CCGNet) [32]. Although the first two methods (i.e., HBP and MEP) have been tested and validated on various APIs [33,63,64,82,83,84,85,86,87,88], the performance of the CCGNet strategy has yet to be put to the test for a specific API, with a relatively wide range of reported experimental trials. The HBP method expresses the likelihood of cocrystal formation in terms of a multicomponent score parameter, which is calculated by subtracting the propensity of the best hetero-interaction between components in a hypothetical cocrystal from the propensity of the best homo-interaction between API or coformer molecules [64]. In the MEP approach, the values of the local maxima and minima on the molecular electrostatic potential surfaces of an API and coformer are identified and further used to calculate α (hydrogen bond-donating sites) and β (hydrogen bond-accepting sites) descriptors for each conceivable interaction site [22]. Then, these values are coupled together in descending order to evaluate the total interaction energy between two API molecules, two coformer molecules, and the combination of an API and coformer molecules. The difference in the obtained pairing energies for a hypothetical API–coformer crystal and the individual compounds provided a measure (in terms of ΔE) to rank the coformers from the most to the least energetically favorable and allowed for the calculation of the energy gain upon cocrystal formation. As a result, the higher the predicted value of ΔE, the greater the likelihood of cocrystal formation. Two variations of the MEP screening method were tested in this work (method A and method B). Method A implies a standard technique for calculating cocrystal energies, in which the cross product of the α and β values for API and coformer molecules is used. Method B has been proposed by Ahmadi et al. [33], who recommended computing the cocrystallization energy utilizing hierarchical mapping of initially combined and sorted α and β lists of a coformer and API. This technique is likely to be relevant for a large variety of CBZ-based cocrystals, where the carboxamide cyclic dimer of CBZ molecules is preserved while coformer molecules are attached to the dimer. The CCGNet method ranks a pair of coformers based on the received predictive score values, with the higher positive values indicating a greater possibility of favorable cocrystallization [32].

The resulting score parameters obtained from the different methods of virtual screening, along with coformer names, CSD Refcodes, and relevant references, are provided in Appendix A. The receiver operator characteristic (ROC) curves were used to compare the performances of the various virtual screening models (Figure 2). A ROC curve depicts the true positive (TP) vs. false positive (FP) prediction rate for a binary classifier when its discrimination threshold is increased from low to high. A random estimate would result in a position along the diagonal dashed line (also known as the “line of no-discrimination”) that runs from the bottom left to the top right corners (Figure 2). The likelihood that a classifier would score a randomly selected positive instance higher than a randomly selected negative one is determined by the area under the ROC curve (AUC), which assesses the model’s overall performance. Thus, the AUC should always be higher than 0.5, indicating that the model is better than random selection. The resulting confusion matrices calculated from the predictions of the tested models on the CBZ-coformer dataset are provided in Appendix A.

A head-to-head comparison of the chosen models indicated that the HBP method possessed the lowest discriminative power to identify which coformer would be more appropriate for cocrystallization with CBZ, as it achieved an AUC value of 0.36, which is even lower than the random guessing result (Figure 2). Its performance on the positive samples (TPR or sensitivity) does not exceed 40% (Table 1), indicating that the HBP approach cannot confidently recognize true positive results and generates a large number of false negatives. The major drawback of HBP is that the multicomponent score values for the majority of tested systems were nearly equal to zero (Appendix A), indicating that the occurrence of the corresponding cocrystals is not completely excluded but does not appear to be advantageous with respect to homodimeric hydrogen bonds in the crystals of parent compounds. As a result, this approach cannot be suggested for the virtual screening of CBZ cocrystals. Concerning the MEP model, method B outperforms method A noticeably, as evidenced by ROC curves with AUC values of 0.72 and 0.65, respectively (Figure 2). Despite the reasonably high sensitivity shown by method A (78.2%, Table 1), it has a poor specificity (28.1%, Table 1), which leads to a low balanced accuracy. Unfortunately, the metric parameters of the prediction performance of method B cannot be compared to that of other commonly used algorithms without a predefined nonzero cutoff separating the predicted positive and negative instances. However, MEP method B can be useful for a preliminary rating of prospective coformers for experimental screening with CBZ, with top-scoring compounds being the most likely cocrystal formers. Even though the CCGNet model was unable to entirely differentiate positive and negative instances for the CBZ-coformer pairs, resulting in a prediction performance similar to MEP (in terms of AUC), it still achieved 80.4% sensitivity and 46.9% specificity, which is much better than that of HBP and MEP method A (Table 1). It should be noted that in the original study, the CCGNet method demonstrated 100% accuracy for a small test set of CBZ combinations with 22 coformers [32]. It is evident, however, that an enlargement of the external list of the experimentally supported positive and negative coformers for CBZ led to a significant decline in the model’s performance, highlighting the need for additional adjustments to the approach for pharmaceutical cocrystals. Nevertheless, the CCGNet model does not require time-consuming quantum mechanics calculations (as in MEP) or data gathering from CSD (as in HPB) and, in its default mode, performs reasonably well for the CBZ validation set. Despite some limitations, this approach can be successfully applied to search for new cocrystals of CBZ, with the caveat that several promising candidates may receive negative scores and appear at the bottom of the list. In particular, CCGNet correctly predicted that all three isomers of acetamidobenzoic acid are expected to form cocrystals with CBZ, whereas the results of MEP methods A and B were inconclusive.

### 3.3. Crystal Structure Analysis and Intermolecular Interaction Energies in [CBZ + 3AcAmBA] (1:1) and [CBZ + 4AcAmBA] (1:1)

Good-quality single crystals and bulk materials for the [CBZ + 3AcAmBA] (1:1) and [CBZ + 4AcAmBA] (1:1) cocrystals were produced using conventional solution crystallization techniques. However, all attempts to obtain the cocrystal between CBZ and 2AcAmBA via solution-based methods appeared to be unsuccessful, resulting in simple physical mixtures of either the starting material or CBZ and the previously unknown 2AcAmBA ethyl acetate solvate (Appendix A).

Both [CBZ + 3AcAmBA] (1:1) and [CBZ + 4AcAmBA] (1:1) crystallize into a monoclinic cell with one CBZ molecule and one coformer molecule in the asymmetric unit. From Figure 3 and Appendix A, one can see that the cocrystals with 3AcAmBA and 4AcAmBA show identical three-dimensional hydrogen bond networks [89] and similar packing. However, the [CBZ + 3AcAmBA] (1:1) and [CBZ + 4AcAmBA] (1:1) cocrystals are not isostructural, as Crystal Packing Similarity analysis [90] showed. The QTAIMC analysis was performed on the relaxed structures of the studied cocrystals in order to quantify and rank the non-covalent interactions and to discuss their role in the packing architecture.

The third rule of hydrogen bonding proposed by M.C. Etter states that “the best proton donors and acceptors that remain after intramolecular hydrogen bond formation form intermolecular hydrogen bonds with one another” [91]. In the MEP analysis, the ranking of donor/acceptor sites is based on the interaction site parameters derived from the electrostatic potential value at the minimum/maximum on the isosurface, allowing one to suggest the most probable pattern of hydrogen bonding in crystal. For multicomponent crystals, the donors and acceptors of all constituent molecules are united in a single ranked list.

The strongest non-covalent interactions in the cocrystals of carbamazepine with 3AcAmBA and 4AcAmBA are O1–H1⋯O3 hydrogen bonds (Figure 3), which are formed between the second-best donor and acceptor sites in the dimer, namely, the O–H_(acid)_ and O=C_(amide)_ groups of AcAmBA (Figure 4, Appendix A). These H-bonds unite the coformer molecules into infinite zigzag chains of C(9) topology in the commonly accepted graph set notation [92] for 3AcAmBA and C(10) topology for 4AcAmBA (Figure 3). The energies of these interactions are estimated by QTAIMC at 42.2 and 45.4 kJ·mol^−1^, respectively. The second strongest non-covalent interaction in both crystals is the heteromolecular N1–H10⋯O11 hydrogen bond with energy equal to 28.5 kJ·mol^−1^ in [CBZ + 3AcAmBA] and 27.0 kJ·mol^−1^ in [CBZ + 4AcAmBA] that forms between the best donor and the best acceptor in the crystal (Figure 4), i.e., the N–H_(amide)_ group of the coformer and the O=C_(amide)_ group of CBZ (Figure 3). It is worth mentioning that the two strongest interactions in both cocrystals obey the third Etter’s rule [91], as the best/second-best H-bond donor is paired with the acceptor of the same rank.

Note that the expected acid-amide heterosynthon [93] is not formed in any of the studied crystals, which is likely caused by the competition between the hydrogen bonding sites in CBZ and a coformer. Instead, the CBZ molecules form centrosymmetric $R_{2}^{2}\left(8\right)$ amide dimers via N11–H11⋯O11 hydrogen bonds with E_int_ = 24–27 kJ·mol^−1^ (the third strongest non-covalent interactions in crystals) that link the best acceptor (O=C_(amide)_ in CBZ) and third-best donor (N–H_(amide)_ in CBZ) (Figure 3). The remaining N–H_(amide)_ group in the CBZ molecule is involved in a weak (17–21 kJ·mol^−1^) N11–H12⋯O2 hydrogen bond with an O=C_(acid)_ fragment of the coformer (the fourth-best acceptor in the crystal). The centrosymmetric amide dimers of CBZ are packed along the crystallographic *c* axis via C–H⋯π and π⋯π interactions, forming the B2” dimeric packing motif in the notation of CBZ supramolecular constructs proposed in the paper by Gelbrich et al. [94] Along the *a*-axis, the CBZ dimers related by translation are held together by a pair of weak C27–H27⋯O11 and C17–H17⋯N11 H-bonds in the case of [CBZ + 4AcAmBA] (1:1) (Appendix A) and a single C27–H27⋯N11 bond in [CBZ + 3AcAmBA] (1:1). In the cocrystal with 4AcAmBA, the energy of chain motifs formed by C–H⋯O/N contacts is comparable with that of the hydrogen-bonded CBZ dimers (22.0 vs. 26.5 kJ·mol^−1^) due to weak stabilizing dihydrogen bonds (not shown in Appendix A). The remaining acceptor sites of both components participate in multiple non-classical C–H⋯O and C–H⋯N hydrogen bonds with E_int_ values within 9–16 kJ mol^−1^ that unite the H-bonded sheets into 3D networks.

Further analysis of a subset of the CBZ cocrystals (1:1 molar ratio) with a variety of the benzoic acid derivatives (Figure 4) revealed the prevalence of an acid−amide heterosynthon in the crystal structures unless competing H-bond donors exist in the coformer’s molecule. The formation of a heterosynthon in the CBZ cocrystals with benzoic, salicylic, 2,6-dihydroxybenzoic, acetylsalicylic, and 3,5-dinitrobenzoic acids is advantageous from both electrostatic and geometric perspectives. Since the only available strongest H-bond donor in the coformer molecule (O–H_(acid)_) is coupled with the strongest H-bond acceptor in CBZ (O=C_(amide)_) (Figure 4), a metric complementarity between the carboxylic and amide fragments facilitates dimeric binding even when stronger H-bond acceptors are present in the molecular structure of a coformer (e.g., salicylic, 2,6-dihydroxybenzoic, acetylsalicylic, and 3,5-dinitrobenzoic acids) (Appendix A). The acid−amide heterosynthon is not observed in either polymorph of the CBZ cocrystal with 4-hydroxybenzoic acid. In these structures, the best donor of the acid molecule (according to MEP values, Figure 4), the para-hydroxyl group (O–H_(ar)_), is involved in hydrogen bonding with the best CBZ acceptor (O=C_(amide)_), leaving the second-best donor and acceptor to form a conventional carboxylic homodimer (Appendix A). Similar hydrogen bond patterns can be found in the CBZ–2,4-dihydroxybenzoic acid cocrystal, albeit the presence of the hydroxyl group in the ortho position increases the MEP energy value at O–H_(acid)_ of the carboxylic group, making it more competitive (Figure 4 and Appendix A). Both hydroxyl groups (O–H_(acid)_ and O–H_(ar)_) available for intermolecular interactions in 2,3-dihydroxybenzoic acid have similar energy levels at the MEP maxima (Figure 4). As a result, both -OH moieties compete for the same best CBZ acceptor (O=C_(amide)_), leading to a complex hydrogen bond network where the acid−amide and O–H_(ar)_⋯O=C_(amide)_ heterosynthons coexist within the same crystalline environment (Appendix A). In the CBZ-2,5-dihydroxybenzoic acid cocrystal, the amide group carbonyl oxygen of CBZ also acts as an acceptor for two hydrogen bonds from the competitive O–H_(acid)_ and O–H_(ar)_ groups (Figure 4) of two distinct coformer molecules. However, neither conventional hetero- nor homo-dimeric synthons are seen in the crystal in this case. Instead, the amide group of CBZ forms a rare $R_{3}^{2}\left(8\right)$ synthon through hydrogen bonding with the carbonyl oxygens of the adjacent acid and CBZ molecules (Appendix A). Since the described hydrogen bond network is non-typical and is not observed in other known cocrystals of the drug with benzoic acid derivatives, it appears likely that the CBZ-2,5-dihydroxybenzoic acid cocrystal may be prone to polymorphism. In the case of CBZ cocrystals containing 4-aminobenzoic and 4-amino-2-hydroxybenzoic acids (in a 1:1 molar ratio), changes in MEP distribution amongst coformers have a considerable impact on the relative strength of the carboxylic group donor site (O–H_(acid)_) (Figure 4). As a result, the hydrogen bonding pattern in the CBZ-4-aminobenzoic acid cocrystal is comparable to that found in the structure of CBZ-4-hydroxybenzoic acid (metastable form), with both amide-amide and acid-acid homodimers produced (Appendix A). Due to the impact of the *ortho* hydroxyl group, the MEP value at the carboxylic hydrogen in 4-amino-2-hydroxybenzoic acid is observed to be noticeably larger than that of the amine hydrogens. Hence, the best donors and acceptors in the CBZ and coformer molecules are paired to form the acid-amide heterosynthon, while the para amino group, the second strongest H-bond donor in 4-amino-2-hydroxybenzoic acid, is involved in H-bonding with the acid’s second-best acceptor, the *ortho* hydroxyl group (Appendix A).

### 3.4. Formation Thermodynamics of the Carbamazepine Cocrystals

It is known that the relative stability of a cocrystal with respect to a physical mixture of the parent constituents at a given temperature can be quantitatively expressed in terms of the free-energy change, Δ_form_G, of the cocrystallization reaction [37,39,40,45]. Analyzing the temperature dependence of the Gibbs formation energy can provide further information on the enthalpy and entropy contributions to the driving force of the process as well as help to estimate the thermodynamic transition temperature, which determines the stability domains of a cocrystal and its individual constituents [40,43]. The experimental assessment of Δ_form_G is based on solubility data for the starting molecules and the corresponding cocrystal in the same solvent at a predetermined temperature using the equation below:(3)$$\Delta_{\mathrm{form}}G=-\mathrm{RTln}\left(\frac{a_{p,A}^{x}\times a_{p,B}^{y}}{a_{\mathrm{cc},A}^{x}\times a_{\mathrm{cc},B}^{y}}\right)=-\mathrm{RTln}\left(\frac{a_{p,A}^{x}\times a_{p,B}^{y}}{K_{\mathrm{sp}}}\right){=-\mathrm{RTlnK}}_{f},$$
where a_p,A_ and a_p,B_ represent the activities of pure A and B compounds in a saturated solution; a_cc,A_ and a_cc,B_ are the activities of the cocrystal components in a solution, in equilibrium with pure cocrystal; K_sp_ is the solubility product of a cocrystal; T is the temperature; x and y are the stoichiometric coefficients; and K_f_ is an equilibrium constant at the experimental temperature.

Using the van’t Hoff expression, the enthalpy of the cocrystallization process, Δ_form_H, may be estimated by evaluating the K_f_ values at various temperatures (assuming that Δ_form_H is a constant within the selected temperature range) [42,44,46,95]:(4)$$\frac{{\mathrm{dlnK}}_{f}}{d(1/T)}=-\frac{\Delta_{\mathrm{form}}H}{R}$$

The entropy of cocrystal formation, Δ_form_S, can be estimated from
(5)$$\Delta_{\mathrm{form}}S=\frac{\Delta_{\mathrm{form}}H-\Delta_{\mathrm{form}}G}{T}$$

Even though the activities in Equation (3) are usually approximated by the corresponding molar concentration while neglecting the activity coefficients, a more valid approximation implies that the activity coefficients for the two solute species are constant along the cocrystal solubility line and equal to the activity coefficients of the pure components in the saturated solutions in the same solvent and temperature [96]. Hence, similar to the Gibbs free energy difference between the polymorphs [97], the solvent used for solubility determination has a minor impact on the resulting value of Δ_form_G unless the activity coefficients are strongly dependent on the solute concentration [44]. Appendix A contains the solubility values of CBZ, coformers, and the CBZ cocrystals with emodin, paeonol, nicotinamide, and glutaric acid in various organic solvents. It is evident that the calculated Δ_form_G parameters (using Equation (3)) for a particular CBZ + coformer system demonstrate good consistency and, therefore, justify the assumption proposed above.

The most convenient way to determine the solubility product of a cocrystal is to ensure that the solution and the cocrystal are at thermodynamic equilibrium, i.e., that the cocrystal is congruently soluble in the chosen solvent. In this work, acetonitrile was primarily utilized as a medium to carry out solubility experiments. The preliminary solution stability studies conducted for the newly obtained CBZ cocrystals with 2AcAmBA, 3AcAmBA, and 4AcAmBA revealed that [CBZ + 2AcAmBA] (1:1) and [CBZ + 3AcAmBA] (1:1) dissolved incongruently in acetonitrile, while other solid forms were stable in the solvent, as the PXRD analysis of the residual phases evidenced. In the case of [CBZ + 3AcAmBA] (1:1), incongruent solubility of the cocrystal was expected given the high solubility ratio of the components in acetonitrile (Appendix A). This is likely to result in an asymmetric ternary phase diagram with a narrow stability region for the [CBZ + 3AcAmBA] (1:1) cocrystal. Though the solubility ratio between CBZ and 2AcAmBA in acetonitrile was even lower than that measured for CBZ and 4AcAmBA (Appendix A), the [CBZ + 2AcAmBA] (1:1) cocrystal was found to be incongruently soluble in the selected solvent. Similar incongruent behavior of [CBZ + 2AcAmBA] (1:1) was observed in several other tested solvents, including methanol, ethanol, and acetone. It is reasonable to assume that the poor solution stability of the [CBZ + 2AcAmBA] (1:1) cocrystal is not due to the large solubility difference between CBZ and 2AcAmBA but rather because of the low intrinsic thermodynamic stability of the binary solid form caused by the lack of energy gain associated with cocrystal formation, which leads to a higher solubility in all solvents considered. The DSC results, which demonstrated a considerable reduction in the melting point of the [CBZ + 2AcAmBA] (1:1) cocrystal compared to the parent pure components (~40 °C, Appendix A), implicitly corroborate this supposition. Since no suitable solvent for solubility studies of [CBZ + 2AcAmBA] (1:1) could be found, this system was not further explored. The [CBZ + 3AcAmBA] (1:1) was confirmed to dissolve congruently in methanol, where the solubility ratio between the components is relatively low (Appendix A).

The experimental values of the thermodynamic solubility of the cocrystals and parent components as well as the temperature dependences of the K_f_ and Δ_form_G parameters are provided in Appendix A. The PXRD patterns of residual solids after solubility testing are shown in Appendix A. The evaluated thermodynamic functions of the CBZ cocrystal formation at 298.15 K are summarized in Table 2. According to the received negative values of the Gibbs energy, the formation of the cocrystals from the corresponding individual components is a spontaneous process [76]. As shown in Appendix A, the driving force of the cocrystallization process decreases as the temperature rises, indicating that the thermodynamic stability of the cocrystals diminishes. Based on the observed trend in the Gibbs energy change, one can further deduce that the cocrystallization process for the examined cocrystals is mostly enthalpy-driven. In particular, the formation process of the CBZ cocrystals with the 3AcAmBA and 4AcAmBA isomers is characterized by different thermodynamic parameters, with [CBZ + 4AcAmBA] (1:1) being relatively more thermodynamically stable with respect to the physical mixture of the components than [CBZ + 3AcAmBA] (1:1). The observed variations in the formation thermodynamics are likely owing to differences in the energetics of the cocrystals as well as alterations in the thermodynamic properties of the solid coformes, i.e., the parent 4AcAmBA and 3AcAmBA [40].

At the next stage, the obtained thermodynamic data were put into the broader context of the published results for different multicomponent systems, including CBZ cocrystals. The literature survey enabled us to collect experimental thermodynamic information on the free-energy, enthalpy, and entropy changes of the cocrystallization reaction for 22 multicomponent systems (Appendix A). Figure 5 summarizes the overall experimental thermodynamic properties in the form of a diagram, where the entropy of the cocrystal formation is plotted against the corresponding enthalpy.

The diagram is divided into eight sectors, each corresponding to a different ratio of the enthalpy and entropy contributions to the Gibbs energy. The sector is formed by two lines: on the one side, the line corresponding to the zero Δ_form_H or TΔ_form_S -value; on the other side, the bisector of the angles formed at the intersection of the coordinates. Isoenergetic curves of Gibbs energy are marked by the dotted lines. Overall, Figure 5 shows that for the majority of the cocrystals, the formation process is mainly controlled by enthalpy (sector D), while the entropy-driven reactions (sector A) were seen in just 23% of the systems (5 out of 22 systems). Although this result is below the predicted number of ca. 30% [20], the figure is expected to grow as more experimental studies on the thermodynamics of the cocrystallization reactions are conducted. It is evident that the thermodynamic parameters are not evenly spread across the diagram. The reported CBZ cocrystals belong to the most densely populated cluster of the enthalpically favored systems, which are confined within relatively narrow boundaries in terms of the Δ_form_H and TΔ_form_S values (from 0 to −20 kJ·mol^−1^). Due to the small number of experimental data points, it is uncertain whether cocrystals 6 and 8 are outliers or the seeds of a separate cluster of systems. It is also worth noting that, whereas enthalpy plays a decisive role in the stabilization of multicomponent crystals located in sector D, the entropy term is also not negligible. In fact, only a handful of the reported cocrystals are characterized by TΔ_form_S ≈ 0 (11, 12, 13, and 14), indicating that these systems should be regarded as exceptions rather than conventional cases. Hence, assessing the relative stability of a cocrystal based solely on static lattice energy calculations [27,47,48] may lead to inaccurate conclusions, since the potential energy gain upon cocrystallization may be offset by a large and opposing entropy contribution, resulting in a relatively small value of the Gibbs energy and low thermodynamic stability. Therefore, further experimental studies on the formation thermodynamics of multicomponent crystals are required not only to rationalize and generalize the structure-energy relationships in these systems but also to provide the thermodynamic grounds for the development of reliable, free energy-oriented virtual screening models.

### 3.5. Stability and Solubility of Carbamazepine Cocrystals in Aqueous Media

The stability of a cocrystal in aqueous solution under stoichiometric conditions is an important characteristic that may have a profound effect on the supersaturated dissolution profile of an API and in vivo performance [98,99,100]. It is widely accepted that the solution stability and the intrinsic solubility of a cocrystal depend on several parameters, including the strength of intermolecular interactions in the cocrystal (lattice energy), the API-to-coformer solubility ratio, and the cocrystal stoichiometry [101]. Rodrguez-Hornedo et al. proposed that the ratio of solution concentrations of cocrystal components at the eutectic point is a constant (referred to as the eutectic constant, or K_eu_) that governs the cocrystal’s thermodynamic stability relative to a drug under given conditions [7]. A cocrystal with a 1:1 molar ratio is considered to be thermodynamically stable if its K_eu_ value does not exceed one [102]. Given that the CBZ cocrystals with the isomers of acetamidobenzoic acid described in this work appear to have the same stoichiometry, their relative stability in water is expected to be mainly determined by the solubility difference between the coformers as well as the packing efficiency of the corresponding cocrystals. According to literature data [103], the aqueous solubility values (in a pH 2.0 buffer) for 3AcAmBA and 4AcAmBA are closely comparable, with 2AcAmBA being slightly more soluble. The compound with the highest intrinsic solubility in water is 2-acetamidobenzoic acid [103]. All the coformers were found to be 4.5–12.0 times more soluble than carbamazepine hydrate under acidic conditions (pH 2.0) and at 310.2 K ((8.30 ± 0.02)·10^−4^ mol·l^−1^). Therefore, the equilibrium solubility of the CBZ cocrystals was determined at the eutectic point. The PXRD analysis of the equilibrium solid samples, indicating the presence of both drug and cocrystal phases, is provided in the Appendix A. The obtained experimental K_eu_ values for the cocrystals clearly highlighted the difference in thermodynamic stability of the systems under the studied conditions (Table 3). Even though the [CBZ + 3AcAmBA] (1:1) and [CBZ + 4AcAmBA] (1:1) cocrystals have analogous crystal structures and contain coformers with similar solubilities, slight variations in packing arrangements and thermodynamic stability make the [CBZ + 4AcAmBA] (1:1) solid form less susceptible to dissociation during dissolution. It is generally accepted that a coformer’s intrinsic solubility should be at least 10 times higher than that of a drug to have a meaningful impact on the solubility advantage of the resulting cocrystal [7,104]. The [CBZ + 2AcAmBA] (1:1) and [CBZ + 4AcAmBA] (1:1) cocrystals were found to obey this empirical rule, though the [CBZ + 3AcAmBA] (1:1) was a notable exception, indicating a complex interplay between solid-state and solute-solvent interactions for a specific class of cocrystals with the API-to-coformer solubility ratio less than 10 [105,106]. In the case of cocrystals with highly soluble coformers, the increase in solubility is mainly due to decreases in the relative free energy of solvation, while the influence of strengthening of the crystal lattice and a gain in a cocrystal’s solid-state stability is usually insignificant [107]. However, if cocrystals are formed from components with comparable solubilities, the difference in solid-state thermodynamic stability can be crucial, dictating their dissolution behavior. According to the results of the thermodynamic analysis shown above, the [CBZ + 4AcAmBA] (1:1) cocrystal was found to be markedly more thermodynamically stable compared to [CBZ + 3AcAmBA] (1:1), which is assumed to be the main reason for the superior stability of the former solid form in aqueous solution. The increased solubility of the [CBZ + 2AcAmBA] (1:1) cocrystal is likely owing to two factors acting concurrently, namely, the relatively high intrinsic water solubility of the coformer and the low melting point of the cocrystal, and the former is usually associated with a low crystal lattice strength of a material.

## 4. Conclusions

In this study, experimental screening of the novel carbamazepine (CBZ) cocrystals was performed and complemented based on the results of three distinct predictive methods (HBP, MEP, and CCGNet). In addition, the thermodynamic parameters of the formation process of the selected CBZ cocrystals were evaluated and compared with literature data.

The experimental screening was conducted with 17 coformers prior to any calculations and resulted in the formation of three new multicomponent solid forms of CBZ containing isomers of acetamidobenzoic acid (2AcAmBA, 3AcAmBA, and 4AcAmBA). Two of these ([CBZ + 3AcAmBA] (1:1) and [CBZ + 4AcAmBA] (1:1)) were structurally characterized, enabling a detailed quantitative analysis of the major intermolecular interactions in the crystals within the framework of the quantum theory of atoms in molecules and crystals. According to the MEP values of the interaction sites, the formation of the conventional hydrogen bonds between the CBZ and coformer follows Etter’s “best donor-best acceptor” rule. However, the strongest non-covalent interactions in both cocrystals were formed between the second-best donor and second-best acceptor sites, both of which are located in the coformer molecule. Furthermore, the MEP surface analysis was also applied to rationalize the competitive formation of acid–amide heterosynthon and amide-amide homosynthon in the crystal structures of CBZ cocrystals with aromatic carboxylic acids.

The overall performance of the tested virtual screening models for the prediction of experimentally verified outcomes of CBZ combinations with 87 different coformers, was analyzed in terms of the area under the ROC curve values and some other complementary metric parameters. The HBP technique demonstrated the lowest discriminative power in determining which coformer would be better suited for cocrystallization with CBZ, with an accuracy even lower than random guessing. Even though the CCGNet and MEP-based methods showed comparable performance in terms of the AUC values, the former resulted in superior specificity and overall accuracy while requiring no time-consuming DFT computations to be carried out. Hence, the CCGNet approach can be recommended for the high-throughput screening of novel CBZ cocrystals.

According to the thermodynamic parameters of cocrystal formation derived from solubility experiments, (i) the cocrystallization reaction between CBZ and the considered coformers (3AcAmBA and 4AcAmBA) is mainly controlled by enthalpy, and (ii) the [CBZ + 4AcAmBA] (1:1) cocrystal is found to be more thermodynamically stable compared to the [CBZ + 3AcAmBA] (1:1). This difference in thermodynamic stability is likely responsible for the superior stability of the [CBZ + 4AcAmBA] (1:1) cocrystal in aqueous solution.

It was also found that most of the experimentally examined cocrystallization reactions, including those reported in this work, are associated with a non-zero entropy change. This fact implies that a theoretical assessment of the relative stability of multicomponent crystals based solely on static lattice energy calculations may lead to a significant overestimation of their true thermodynamic stability.

## Figures and Tables

**Figure 1 pharmaceutics-15-00836-f001:**
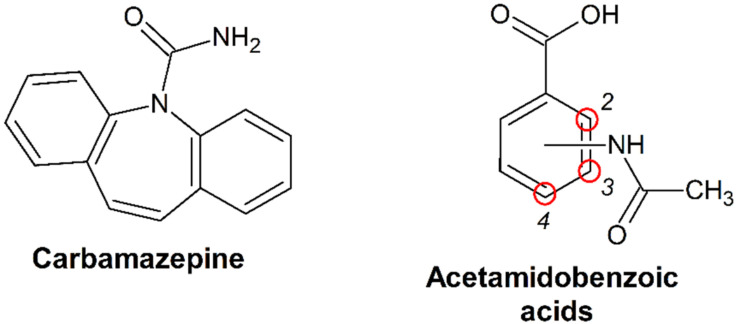
Molecular structures of carbamazepine and isomers of acetamidobenzoic acid studied in this work.

**Figure 2 pharmaceutics-15-00836-f002:**
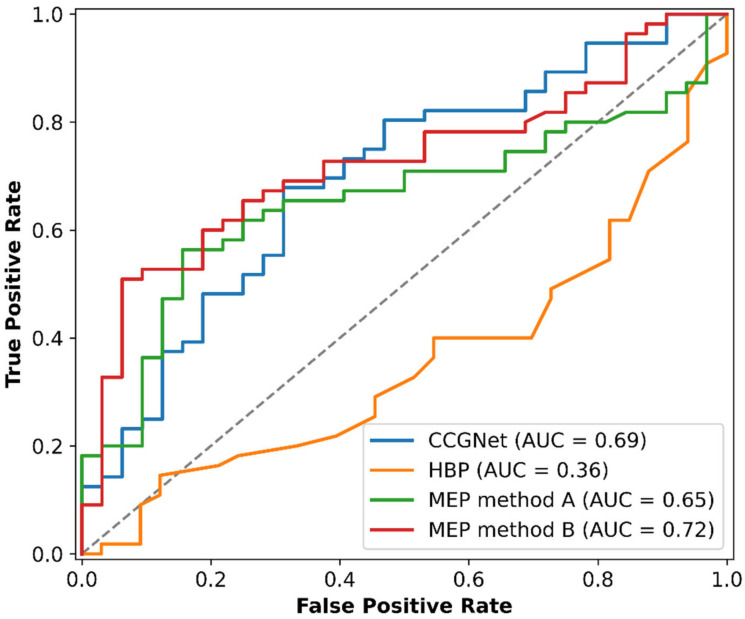
Receiver operating characteristic (ROC) curves and AUC values for four prediction models for the CBZ-coformer dataset.

**Figure 3 pharmaceutics-15-00836-f003:**
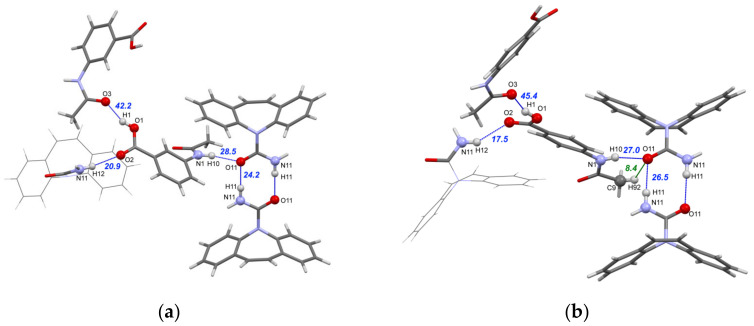
Part of the hydrogen bond network in the [CBZ + 3AcAmBA] (1:1) cocrystal (**a**) and in the [CBZ + 4AcAmBA] (1:1) cocrystal (**b**). The energies denote the interaction energies of conventional (blue dotted lines) and non-conventional H-bonds (green dotted lines), estimated using Equation (2) in kJ·mol^−1^.

**Figure 4 pharmaceutics-15-00836-f004:**
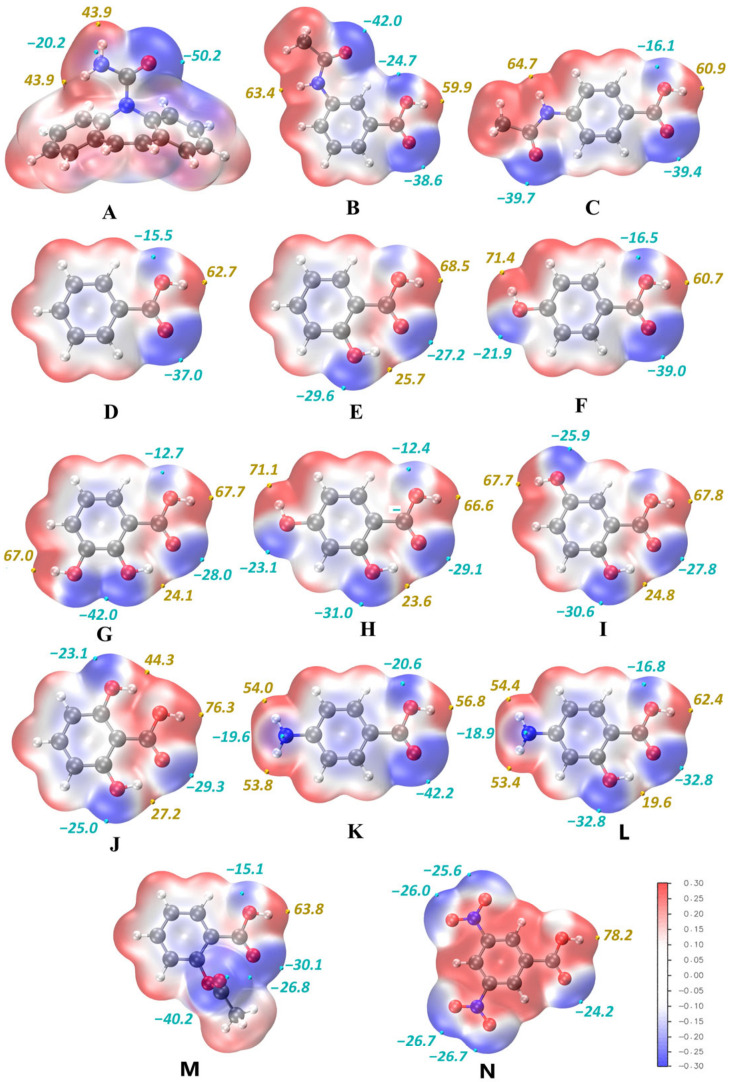
Molecular electrostatic potential mapped over total electron density ρ = 0.002 isosurface for relaxed molecules of CBZ (**A**) and benzoic acid derivatives that form cocrystals with CBZ: 3AcAmBA (**B**), 4AcAmBA (**C**), benzoic acid (**D**), salicylic acid (**E**), 4-hydroxybenzoic acid (**F**), 2,3-dihydroxybenzoic acid (**G**), 2,4-dihydroxybenzoic acid (**H**), 2,5-dihydroxybenzoic acid (**I**), 2,6-dihydroxybenzoic acid (**J**), 4-aminobenzoic acid (**K**), 4-aminosalicylic acid (**L**), acetylsalicylic acid (**M**), 3,5-dinitrobenzoic acid (**N**). The isosurfaces are color-coded in the range between −0.3 a.u. (blue) and +0.3 a.u. (red). The numbers display the selected ESP values of minima (cyan points) and maxima (yellow points) of selected electrostatic potential values in kcal mol^−1^.

**Figure 5 pharmaceutics-15-00836-f005:**
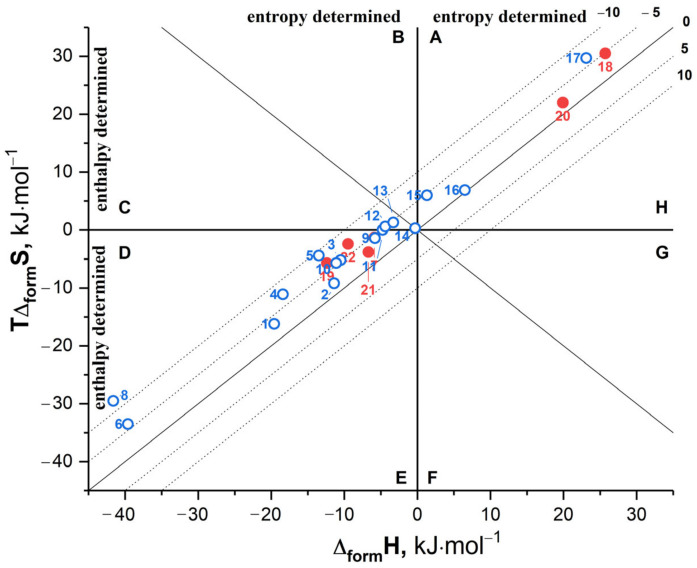
The experimental thermodynamic parameters of cocrystallization processes for 22 systems in terms of entropy vs. enthalpy coordinates. The dotted lines correspond to isoenergetic values of Δ_form_G. The red points indicate CBZ cocrystals, whereas the blue ones correspond to cocrystals of various APIs. The numbering is provided in Appendix A.

**Table 1 pharmaceutics-15-00836-t001:** Prediction performances of the selected models on the CBZ-coformer dataset.

Model	TPR (Sensitivity), %	TNR (Specificity), %	Accuracy, %	Balanced Accuracy, %
HPB	40.0	45.5	42.0	42.7
MEP method A	78.2	28.1	59.8	53.2
CCGNet	80.4	46.9	68.2	63.6

**Table 2 pharmaceutics-15-00836-t002:** Equilibrium constants and thermodynamic functions (in kJ·mol^−1^) of the CBZ cocrystal formation at 298.2K.

Cocrystal	lnK_f_	Δ_form_G	Δ_form_H	T·Δ_form_S
[CBZ + 3AcAmBA] (1:1)	1.17 ± 0.02	−2.91 ± 0.05	−6.7 ± 0.5	−3.8 ± 0.5
[CBZ + 4AcAmBA] (1:1)	2.86 ± 0.03	−7.09 ± 0.08	−9.5 ± 0.6	−2.4 ± 0.6

**Table 3 pharmaceutics-15-00836-t003:** Experimental values of the intrinsic solubility of coformers (S_0_(CF)), eutectic concentrations of CBZ ([CBZ]_eu_) and coformers ([CF]_eu_), calculated values of the eutectic constant (K_eu_), and the solubility of the cocrystals at pH 2.0 and 310.2 K.

Cocrystal	S_0_(CF), ^a^ Mol·L^−1^	[CBZ]_eu_, Mol·L^−1^	[CF]_eu_, Mol·L^−1^	K_eu_ ^b^	S_cc_, ^c^ Mol·L^−1^
[CBZ + 2AcAmBA] (1:1)	11.5·10^−3^	(1.10 ± 0.06)·10^−3^	(7.5 ± 0.1)·10^−3^	6.8 ± 0.4	(2.8 ± 0.1)·10^−3^
[CBZ + 3AcAmBA] (1:1)	4.1·10^−3^	(9.8 ± 0.2)·10^−4^	(1.75 ± 0.08)·10^−3^	1.8 ± 0.1	(1.30 ± 0.03)·10^−3^
[CBZ + 4AcAmBA] (1:1)	4.6·10^−3^	(1.00 ± 0.05)·10^−3^	(6.4 ± 0.2)·10^−4^	0.64 ± 0.04	(0.80 ± 0.02)·10^−3^

^a^ Data taken from [103], ^b^ K_eu_ = [CF]_eu_/[CBZ]_eu_ [108], ^c^${\;S}_{\mathrm{CC}}=\sqrt{{\left[\mathrm{CBZ}\right]}_{\mathrm{eu}}\cdot {\left[\mathrm{CF}\right]}_{\mathrm{eu}}}$.

## Data Availability

The results obtained for all experiments performed are shown in the manuscript and SI, the raw data will be provided upon request.

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
