# Peer review of "Virtual Screening, Structural Analysis, and Formation Thermodynamics of Carbamazepine Cocrystals"

_pharmaceutics, 2023, doi:10.3390/pharmaceutics15030836_

Round 1

Reviewer 1 Report

It is strategically a good idea to select carbamazepine for this work as it is well-known for its multicomponent systems and its susceptibility form different solid forms in the literature.

Machine learning approach already in place to predict the formation of multicomponent systems, however it is still in the developing stage.

It is interesting to see the performance of the CCGNet strategy, and authors used Receiver operating characteristic (ROC) curves to compare the performances of the virtual screening models.  The data shows that CCGNet is better than HBP and MEP method A as well and almost similar to MEP method B. Even though the CCGNet model was unable to entirely differentiate positive and negative instances for the CBZ-coformer pairs, resulting in a prediction performance similar to MEP.

It is interesting to note that the CCGNet does not require longer times to give the predicted result irrespective to some of the other predictive methods in literature.

Also, the thermodynamic parameters of the cocrystallization process are discussed seldom in the literature so it is good to see this information in the manuscript.

Authors have not conducted any solubility study for CBZ, it may be good idea to have this information as well for the better comparison purposes.

Also, please note usually, the low melting cocrystals and metastable forms shows improved solubility, apart from the theoretical explanation that authors provided in this manuscript conclusion.

I can’t see any stability data for the solid forms after the solubility study. It is important to see the stability of these multi-component systems in the given solvent media after completing the solubility to understand how long they are stable. Have authors done any PXRD on these materials? If yes please provide this information in the supporting documents.

Also, authors are claiming that the CBZ+2AcAmBA is a cocrystal without satisfactory evidence. For argument’s sake, if I say this solid form is a salt, how authors can prove with the given data it is not a salt it is a cocrystal? Please note that there are reported salts for CBZ molecule even though it is considered as non-ionizable molecule.

PXRD and DSC data it are not sufficient to prove that the obtained solid form is a cocrystal or a salt. NMR sometimes help to identify the proton transfer, when there is no SCXRD data available. So, please include either NMR data or provide proper justification for calling CBZ+2AcAmBA as cocrystal.

A small comment for ESI: In the crystallography table add Z value (8) for 2AcAmBA ethyl acetate solvate. 

Apart from these minor comments I believe that this a good piece of work that can fit in to the pharmaceutics journal.

Author Response

Response_to_Reviewer_1

It is strategically a good idea to select carbamazepine for this work as it is well-known for its multicomponent systems and its susceptibility form different solid forms in the literature. Machine learning approach already in place to predict the formation of multicomponent systems, however it is still in the developing stage. It is interesting to see the performance of the CCGNet strategy, and authors used Receiver operating characteristic (ROC) curves to compare the performances of the virtual screening models. The data shows that CCGNet is better than HBP and MEP method A as well and almost similar to MEP method B. Even though the CCGNet model was unable to entirely differentiate positive and negative instances for the CBZ-coformer pairs, resulting in a prediction performance similar to MEP. It is interesting to note that the CCGNet does not require longer times to give the predicted result irrespective to some of the other predictive methods in literature. Also, the thermodynamic parameters of the cocrystallization process are discussed seldom in the literature so it is good to see this information in the manuscript.

We would like to thank the Reviewer for carful inspection of the manuscript and valuable comments.

Comment:

Authors have not conducted any solubility study for CBZ, it may be good idea to have this information as well for the better comparison purposes.

Response:

Carbamazepine is a nonionizing drug with a low solubility, the value of which does not depend on the pH of the dissolution medium. Moreover, in an aqueous solution, carbamazepine rapidly converts into its hydrate form, which has lower solubility than the anhydrate form. In the present study, the solubility of the parent carbamazepine (hydrate form) as well as the cocrystals was determined in a pH 2.0 buffer solution. Moreover, carbamazepine rapidly converts into a hydrate form in aqueous solution, which has lower solubility than the anhydrate form. In the present study, the solubility of the parent carbamazepine (hydrate form), as well as the cocrystals, was determined in a pH 2.0 buffer solution. The experimentally determined carbamazepine hydrate solubility in buffer solution at 310.2 K is (8.30±0.02)·10-4 mol·l-1 (the value has been added in the main text), which is consistent with the literature data (García, M. A. et al. (2021). Journal of Pharmaceutical Sciences, 110(5), 1935-1947).

In contrast to carbamazepine, the solubility of acetamidobenzoic acid isomers depends strongly on the pH of the buffer solution. According to the pKa values (3.63 for 2AcAmBA, 3.99 for 3AcAmBA, and 4.27 for 4AcAmBA) and acid-base equilibrium diagram (Manin, A. N., Drozd, K. V., & Perlovich, G. L. (2022). Journal of Molecular Liquids, 347, 118320), the pH range between 1 and 2 has the highest percentage of neutral forms of compounds. Therefore, all solubility experiments were conducted in hydrochloric buffer solution (pH 2.0), where both components (carbamazepine and i-AcAmBA) are predominantly nonionized.

Comment:

Also, please note usually, the low melting cocrystals and metastable forms shows improved solubility, apart from the theoretical explanation that authors provided in this manuscript conclusion.

Response:

We agree the relevant note was introduced in the section 3.5.

Comment:

I can’t see any stability data for the solid forms after the solubility study. It is important to see the stability of these multi-component systems in the given solvent media after completing the solubility to understand how long they are stable. Have authors done any PXRD on these materials? If yes please provide this information in the supporting documents.

Response:

The PXRD method was used to monitor the stability of the solid forms following the solubility investigation in the specified solvent medium (organic solvents or the buffer solution). The [CBZ+3AcAmBA] or [CBZ+4AcAmBA] cocrystals are stable in the selected organic solvent, according to the results of PXRD examination of the remaining solids (methanol or acetonitrile). The experimental PXRD patterns of the solids after the solubility test have been added to the supporting information (Figure S9). The PXRD patterns of the solid phases following solubility tests in buffer solution, demonstrating the existence of carbamazepine hydrate and cocrystal phases, have further been provided in the Supporting Information section (Figure S10).

Comment:

Also, authors are claiming that the CBZ+2AcAmBA is a cocrystal without satisfactory evidence. For argument’s sake, if I say this solid form is a salt, how authors can prove with the given data it is not a salt it is a cocrystal? Please note that there are reported salts for CBZ molecule even though it is considered as non-ionizable molecule.

Response:

Indeed, there have been several studies reporting salt forms of carbamazepine (Perumalla, S. R., & Sun, C. C. (2012). Chemistry–A European Journal, 18(21), 6462-6464; Eberlin, A. R., Eddleston, M. D., & Frampton, C. S. (2013). Acta Crystallographica Section C, 69(11), 1260-1266; Buist, A. R. et al. (2013) Crystal growth & design, 13(11), 5121-5127; Buist, A. R. et al. (2015) Crystal Growth & Design, 15(12), 5955-5962). These works revealed that protonation of the carbonyl group of CBZ can only be achieved via reactions with either concentrated strong mineral acids (HCl, HBr) or sulfonic acids (methanesulfonic, benzenesulfonic acids, etc.). In contrast, the isomers of acetamidobenzoic acid used in this study have pKa values of 3.63, 3.99, and 4.27 for the 2, 3, and 4 isomers, respectively (Manin, A. N., Drozd, K. V., & Perlovich, G. L. (2022). Journal of Molecular Liquids, 347, 118320), which is nowhere near the strength of hydrochloric acid or sulfonic acids. In addition, multiple examples of CBZ multicomponent crystals with various organic acids of comparable and greater strength indicate that the drug only forms cocrystals, whereas the salt forms of CBZ are highly uncommon and limited to the cases mentioned above. Even though we were not able to obtain single crystals for the [CBZ+2AcAmBA] (1:1) solid form and determine its crystal structure, we can confidently claim that the resulting multicomponent crystal is indeed a cocrystal.

Comment:

PXRD and DSC data it are not sufficient to prove that the obtained solid form is a cocrystal or a salt. NMR sometimes help to identify the proton transfer, when there is no SCXRD data available. So, please include either NMR data or provide proper justification for calling CBZ+2AcAmBA as cocrystal.

Response:

The pKa values of carbamazepine available in the literature (pKa of the amide group ≈13.9) [Jones, O. A. H., Voulvoulis, N., & Lester, J. N. (2002). Aquatic environmental assessment of the top 25 English prescription pharmaceuticals. Water research, 36(20), 5013-5022. DOI: 10.1016/S0043-1354(02)00227-0; Oppel, J., Broll, G., Löffler, D., Meller, M., Römbke, J., & Ternes, T. (2004). Science of the total environment, 328(1-3), 265-273. DOI: 10.1016/j.scitotenv.2004.08.005] imply that CBZ exists in the neutral form in the wide pH range. ChemAxon’s Marvin software predicts the pKa value of the protonated amide group of CBZ as -3.75, which is 7.38 units lower than that of the carboxylic group of 2AcAmBA in water at 25°C (3.63, data from [Dean, J. A. Lange’s Handbook of Chemistry, 15th Ed.; McGraw-Hill, Inc.: 1999]). In the compilation by A. Cruz-Cabeza containing 6465 crystals from the CSD formed by acid-base pairs [Cruz-Cabeza, A. J. (2012). CrystEngComm, 14(20), 6362-6365. DOI: 10.1039/C2CE26055G], the cocrystals form in 100% of systems with ΔpKa < -4.

The protonation of the amide oxygen observed in some CBZ crystal forms from the CSD is currently the only option to obtain salt forms of CBZ as a cation. However, it requires strong mineral acids (HCl, HBr) or organic sulfonic acids (methanesulfonic acid, benzenesulfonic acid etc.) to ionize the carbamazepine molecule. [Buist, A. R., Kennedy, A. R., Shankland, K., Shankland, N., & Spillman, M. J. (2013). Crystal growth & design, 13(11), 5121-5127. DOI: 10.1021/cg401341y]. The weakest acid forming salts with CBZ (methanesulfonic acid) [Eberlin, A.R., Eddleston, M.D. and Frampton, C.S. (2013), Acta Cryst. C, 69: 1260-1266. DOI: 10.1107/S010827011302859X] has the pKa of -2 compared to 3.63 of 2AcAmBA. From the carboxylic acid family, a notable example is the cocrystal of CBZ with 2,6-dihydroxybenzoic acid, which pKa is 1.22 (CSD refcode JANZOR). Therefore, the salt formation between CBZ and 2AcAmBA seems implausible.

The NMR data can be used to monitor salt formation in acid-base complexes [e.g. Kim, H., Babu, C. R., & Burgess, D. J. (2013). International journal of pharmaceutics, 448(1), 123-131. DOI: 10.1016/j.ijpharm.2013.03.040]. However, this technique is less informative in distinguishing salts from cocrystals, since the pKa and ionization state of components in crystal and in the solution might differ significantly due to differences in dielectric permittivity and molecular environment.

Comment:

A small comment for ESI: In the crystallography table add Z value (8) for 2AcAmBA ethyl acetate solvate. 

Response:

We thank the Reviewer for a detailed inspection of the Supporting Information. The Z-value for the 2AcAmBA ethyl acetate solvate was inserted in Table S3.

Reviewer 2 Report

In this paper, the authors carried out co-crystal studies on carbamazepine with the positional isomers of acetamidobenzoic acid, including the prediction, preparation, characterization and evaluation. This is a complete research paper, which has important reference value for related researches in this field. However, before it can be published in this journal, there are two questions for the authors to respond to

(1) The authors need to introduce the properties of acetaminobenzoic acid and the purpose to select them as CCF to carry out carbamazepine co-crystal research.

(2) Was the carbamazepine solubility in different solutions with other pH values investigated?Whether the solubility has been improved?

Author Response

Response_to_Reviewer_2

In this paper, the authors carried out co-crystal studies on carbamazepine with the positional isomers of acetamidobenzoic acid, including the prediction, preparation, characterization and evaluation. This is a complete research paper, which has important reference value for related researches in this field. However, before it can be published in this journal, there are two questions for the authors to respond to

Comment:

 (1) The authors need to introduce the properties of acetaminobenzoic acid and the purpose to select them as CCF to carry out carbamazepine co-crystal research.

Response:

We have to stress that carbamazepine is an extremely well-studied compound in terms of cocrystallization, with dozens of multicomponent solid forms identified. One of the main goals of this work was to extend the solid-state landscape of the drug through cocrystallization with a particular set of coformers that have never been tested against CBZ previously but, at the same time, contain functional groups that are possibly advantageous for hydrogen bonding with the CBZ carboxamide moiety (i.e. carboxyl, hydroxy, amino, carboxamide, acetamido groups and their combinations). According to a survey of the CSD, the cocrystal formation between CBZ and compounds bearing the acetamido group does not seem to have been thoroughly investigated, as no instances of such multicomponent crystals can be found. The single reported attempt to combine carbamazepine with acetaminophen (4-acetamidophenol) failed to produce a new solid phase and resulted in a simple physical mixture (Roca-Paixão, L. et al. (2019). CrystEngComm, 21(45), 6991-7001). Therefore, investigation of new multicomponent crystals of CBZ with isomers of acetamidobenzoic acids is of great importance in terms of the fundamental understanding of the cocrystallization process, as the identification of the key supramolecular synthons responsible for the stabilization of the crystalline environment is the basis of crystal engineering. Further exploration of the rich structural landscape of this drug may help to find and rationalize important structure-property relationships between the packing features, strength and patterns of intermolecular interactions in solid forms, and various pharmaceutically relevant physicochemical parameters.

Although acetamidobenzoic acids are rarely used as coformers in the cocrystallization trials, 4-acetamidobenzoic acid has been employed to obtain new pharmaceutical cocrystals with different drugs, such as linogliride (Chrzanowski, F. A., & Ahmad, K. (2017). Drug Development and Industrial Pharmacy, 43(3), 421-431), nicotine (Dull GM, Carr A, Sharp E. PCT Int. Appl. (2015). WO 2015183801 A1. Dec 03, 2015), and upadacitinib (Mattei A. PCT Int. Appl. (2022). WO 2022217257 A1. Oct 13, 2022). In addition, in the human body, 4-acetamidobenzoic acid (Acedoben) is recognized as a metabolite of such drugs as 4-aminobenzoic acid (Nortje, C. et al. (2015). Bioanalysis, 7(10), 1211-1224) and benzocaine (Szoke, A. et al. (1997). Journal of pharmaceutical and biomedical analysis, 16(1), 69-75). The compound is also a component of several pharmaceutical formulations, including isoprinosine (Sliva, J. et al. (2019). Advances in therapy, 36(8), 1878-1905; Beran, J. et al. (2021). Viruses, 13(11), 2246) and deanol acetamidobenzoate (Wermuth, C. G., & Stahl, P. H. (2002). Selected procedures for the preparation of pharmaceutically acceptable salts. in Handbook of pharmaceutical salts: properties, selection, and use, pp. 249-263). Therefore, the cocrystal comprised of carbamazepine and 4-aminobenzoic acid may be considered a drug-drug product.

Part of the provided above assertions were added to the Introduction section to justify selection of the acetamidobenzoic acids for cocrystallization with CBZ.

Comment:

 (2) Was the carbamazepine solubility in different solutions with other pH values investigated?Whether the solubility has been improved?

Response:

Carbamazepine is a nonionizing drug with a low solubility, the value of which does not depend on the pH of the dissolution medium. Moreover, in an aqueous solution, carbamazepine rapidly converts into its hydrate form, which has lower solubility than the anhydrate form. In the present study, the solubility of the parent carbamazepine (hydrate form) as well as the cocrystals was determined in a pH 2.0 buffer solution. Moreover, carbamazepine rapidly converts into a hydrate form in aqueous solution, which has lower solubility than the anhydrate form. In the present study, the solubility of the parent carbamazepine (hydrate form), as well as the cocrystals, was determined in a pH 2.0 buffer solution. The experimentally determined carbamazepine hydrate solubility in buffer solution at 310.2 K is (8.30±0.02)·10-4 mol·l-1 (the value has been added in the main text), which is consistent with the literature data (García, M. A. et al. (2021). Journal of Pharmaceutical Sciences, 110(5), 1935-1947).

In contrast to carbamazepine, the solubility of acetamidobenzoic acid isomers depends strongly on the pH of the buffer solution. According to the pKa values (3.63 for 2AcAmBA, 3.99 for 3AcAmBA, and 4.27 for 4AcAmBA) and acid-base equilibrium diagram (Manin, A. N., Drozd, K. V., & Perlovich, G. L. (2022). Journal of Molecular Liquids, 347, 118320), the pH range between 1 and 2 has the highest percentage of neutral forms of compounds. Therefore, all solubility experiments were conducted in hydrochloric buffer solution (pH 2.0), where both components (carbamazepine and i-AcAmBA) are predominantly nonionized.

Reviewer 3 Report

Comments:

This study extended the existing set of carbamazepine cocrystals by combining the drug with positional isomers of acetamidobenzoic acid, and their structural and energetic features were determined. Three virtual screening methods were evaluated for predicting the correct cocrystallization outcome, and the formation thermodynamic parameters of the new cocrystals were assessed, with the observed difference in dissolution behavior attributed to variations in their thermodynamic stability. The authors need to address some concerns before it is considered for publication.

1. Line 102, since acetamidobenzoic acid can cause skin, eye, and respiratory irritation, could you elaborate why you chose acetamidobenzoic acid as a co-former in your study?

2. Line 128, in this part, could you also please mention what temperature you used for solution crystallization?

3. Line 157, in this part, TGA should also have been conducted to determine weight loss and the decomposition onset temperature.

4. Line 163, other pH buffers should also have been used for solubility test, for example pH7.0 to mimic oral, duodenum, and small intestine environment.

5. Line 163, dissolution rate of your cocrystals should also have been studied as well. You could either measure accurate or approximate dissolution rate. Here are some references you can use:

1) https://doi.org/10.3390/cryst12020275 (Part 2.2.7. accurate dissolution rate measurement using a USP Type II dissolution test apparatus)

2) https://doi.org/10.1021/acs.cgd.0c01197 (Part 8 in SI, approximate dissolution rate measurement)

6. Could you also provide 1H NMR data of your cocrystals?

7. Could you also use the ΔpKa rule to further confirm the products you obtained are cocrystals instead of salts? See the following references:

1) https://doi.org/10.1021/mp0601345 (Page 335-326)

2) https://doi.org/10.1039/C2CE26055G (Page 6363-6364)

3) https://doi.org/10.1021/acs.cgd.0c00210 (Page 5055-5058)

Author Response

Response_to_Reviewer_3

This study extended the existing set of carbamazepine cocrystals by combining the drug with positional isomers of acetamidobenzoic acid, and their structural and energetic features were determined. Three virtual screening methods were evaluated for predicting the correct cocrystallization outcome, and the formation thermodynamic parameters of the new cocrystals were assessed, with the observed difference in dissolution behavior attributed to variations in their thermodynamic stability. The authors need to address some concerns before it is considered for publication.

Comment:

  1. Line 102, since acetamidobenzoic acid can cause skin, eye, and respiratory irritation, could you elaborate why you chose acetamidobenzoic acid as a co-former in your study?

Response:

First of all, we have to emphasize that this research is fundamental in nature, focusing on expanding the set of multicomponent solid forms of carbamazepine via cocrystallization with a specific set of coformers that have never been tested against CBZ before but contain functional groups that may be advantageous for hydrogen bonding with the CBZ carboxamide moiety (i.e. carboxyl, hydroxy, amino, carboxamide, acetamido groups and combinations thereof). According to a survey of the CSD, the cocrystal formation between CBZ and compounds bearing the acetamido group does not seem to have been thoroughly investigated, as no instances of such multicomponent crystals can be found. The single reported attempt to combine carbamazepine with acetaminophen (4-acetamidophenol) failed to produce a new solid phase and resulted in a simple physical mixture (Roca-Paixão, L. et al. (2019). CrystEngComm, 21(45), 6991-7001). Therefore, investigation of new multicomponent crystals of CBZ with isomers of acetamidobenzoic acids is of great importance in terms of the fundamental understanding of the cocrystallization process, as the identification of the key supramolecular synthons responsible for the stabilization of the crystalline environment is the basis of crystal engineering. Further exploration of the rich structural landscape of this drug may help to find and rationalize important structure-property relationships between the packing features, strength and patterns of intermolecular interactions in solid forms, and various pharmaceutically relevant physicochemical parameters.

Secondarily, although acetamidobenzoic acids are rarely used as coformers in the cocrystallization trials, 4-acetamidobenzoic acid has been employed to obtain new pharmaceutical cocrystals with different drugs, such as linogliride (Chrzanowski, F. A., & Ahmad, K. (2017). Drug Development and Industrial Pharmacy, 43(3), 421-431), nicotine (Dull GM, Carr A, Sharp E. PCT Int. Appl. (2015). WO 2015183801 A1. Dec 03, 2015), and upadacitinib (Mattei A. PCT Int. Appl. (2022). WO 2022217257 A1. Oct 13, 2022). In addition, in the human body, 4-acetamidobenzoic acid (Acedoben) is recognized as a metabolite of such drugs as 4-aminobenzoic acid (Nortje, C. et al. (2015). Bioanalysis, 7(10), 1211-1224) and benzocaine (Szoke, A. et al. (1997). Journal of pharmaceutical and biomedical analysis, 16(1), 69-75). The compound is also a component of several pharmaceutical formulations, including isoprinosine (Sliva, J. et al. (2019). Advances in therapy, 36(8), 1878-1905; Beran, J. et al. (2021). Viruses, 13(11), 2246) and deanol acetamidobenzoate (Wermuth, C. G., & Stahl, P. H. (2002). Selected procedures for the preparation of pharmaceutically acceptable salts. in Handbook of pharmaceutical salts: properties, selection, and use, pp. 249-263). Therefore, the cocrystal comprised of carbamazepine and 4-acetamidobenzoic acid may be considered a drug-drug product.

Part of the provided above assertions were added to the Introduction section to justify selection of the acetamidobenzoic acids for cocrystallization with CBZ.

Comment:

  1. Line 128, in this part, could you also please mention what temperature you used for solution crystallization?

Response:

We agree. The crystallization temperature was specified in the Materials and Methods section (2.3. Solution Crystallization)

Comment:

  1. Line 157, in this part, TGA should also have been conducted to determine weight loss and the decomposition onset temperature.

Response:

Thermogravimetric analysis (TGA) of the carbamazepine crystals has been conducted. The experimental TGA curves have been added to the supporting information section (Figure S3). TGA curves showed no mass loss until melting of the cocrystals, suggesting that the obtained carbamazepine cocrystals were not solvates or hydrates. TGA measurements of these cocrystals showed that there was rapid mass loss after melting temperatures, which was attributed to the degradation of cocrystals.

Comment:

  1. Line 163, other pH buffers should also have been used for solubility test, for example pH≈7.0 to mimic oral, duodenum, and small intestine environment.

Response:

Carbamazepine is a nonionizing drug with a low solubility, the value of which does not depend on the pH of the dissolution medium. Moreover, in an aqueous solution, carbamazepine rapidly converts into its hydrate form, which has lower solubility than the anhydrate form. In the present study, the solubility of the parent carbamazepine (hydrate form) as well as the cocrystals was determined in a pH 2.0 buffer solution. Moreover, carbamazepine rapidly converts into a hydrate form in aqueous solution, which has lower solubility than the anhydrate form. In the present study, the solubility of the parent carbamazepine (hydrate form), as well as the cocrystals, was determined in a pH 2.0 buffer solution. The experimentally determined carbamazepine hydrate solubility in buffer solution at 310.2 K is (8.30±0.02)·10-4 mol·l-1 (the value has been added in the main text), which is consistent with the literature data (García, M. A. et al. (2021). Journal of Pharmaceutical Sciences, 110(5), 1935-1947).

In contrast to carbamazepine, the solubility of acetamidobenzoic acid isomers depends strongly on the pH of the buffer solution. According to the pKa values (3.63 for 2AcAmBA, 3.99 for 3AcAmBA, and 4.27 for 4AcAmBA) and acid-base equilibrium diagram (Manin, A. N., Drozd, K. V., & Perlovich, G. L. (2022). Journal of Molecular Liquids, 347, 118320), the pH range between 1 and 2 has the highest percentage of neutral forms of compounds. Therefore, all solubility experiments were conducted in hydrochloric buffer solution (pH 2.0), where both components (carbamazepine and i-AcAmBA) are predominantly nonionized.

Comment:

  1. Line 163, dissolution rate of your cocrystals should also have been studied as well. You could either measure accurate or approximate dissolution rate. Here are some references you can use:

1) https://doi.org/10.3390/cryst12020275 (Part 2.2.7. accurate dissolution rate measurement using a USP Type II dissolution test apparatus)

2) https://doi.org/10.1021/acs.cgd.0c01197 (Part 8 in SI, approximate dissolution rate measurement)

Response:

We have to stress that the principal goal of solubility studies performed in this paper was to evaluate thermodynamic solubility of the cocrystal in aqueous media and to assess solution stability of the solid forms. Based on the received solubility data and the Keu values, we concluded that the CBZ cocrystals with 2- and 3-acetamidobenzoic acids are more soluble than the carbamazepine hydrate and, therefore, are expected to be thermodynamically unstable in water solution. It has been reported that incongruently soluble CBZ cocrystals undergo rapid and irreversible solution-mediated phase transformation (SMPT) to the CBZ dihydrate during the dissolution process (Omori, M. et al. (2020). Molecular Pharmaceutics, 17(10), 3825-3836; Omori, M. et al. (2020). Journal of Drug Delivery Science and Technology, 56, 101566; Yamashita, H., & Sun, C. C. (2018). Pharmaceutical research, 35, 1-7; Yamashita, H., & Sun, C. C. (2017). CrystEngComm, 19(8), 1156-1159; Omori, M., & Sugano, K. (2021). Crystal Growth & Design, 21(11), 6237-6244). As a result, little or no improvement in dissolution rate and apparent solubility can be observed in the bulk phase. It was argued that the rate of the nucleation and crystallization process of the CBZ hydrate seemed to be higher than the rate of cocrystal dissolution, so that the resulting dissolution curves for CBZ cocrystals and CBZ hydrate in the blank buffer are virtually identical (e.g., see Figure 4 in Omori, M. et al. (2020). Molecular Pharmaceutics, 17(10), 3825-3836). It has been shown that additional formulation techniques, such as the utilization of pharmaceutically acceptable polymers, are required to suppress the nucleation of CBZ in a supersaturated aqueous solution and to unlock the dissolution potential of cocrystals (Omori, M. et al. (2020). Molecular Pharmaceutics, 17(10), 3825-3836; Shigemura, M. et al. (2022). Journal of Drug Delivery Science and Technology, 67, 103029; Omori, M. et al. (2022). Pharmaceutical Research, 1-13). Therefore, the dissolution experiments for the CBZ cocrystals with acetamidobenzoic acids are anticipated to be uninformative due to the mentioned SMPT process unless appropriate precipitation inhibitors are used. The investigation of the dissolution behavior of the CBZ cocrystals in the presence of such additives is a relatively time-consuming task that cannot be accomplished within the time frame allocated for revision of the manuscript. These studies are intended to be performed in our future work pertaining to the dissolution features of CBZ cocrystals.

Comment:

  1. Could you also provide 1H NMR data of your cocrystals? (AV)

Response:

The NMR data can be used to monitor salt formation in acid-base complexes [e.g. Kim, H., Babu, C. R., & Burgess, D. J. (2013). International journal of pharmaceutics, 448(1), 123-131. DOI: 10.1016/j.ijpharm.2013.03.040]. However, this technique is less informative in distinguishing salts from cocrystals, since the pKa and ionization state of components in crystal and in the solution might differ significantly due to differences in dielectric permittivity and molecular environment.

Comment:

  1. Could you also use the ΔpKarule to further confirm the products you obtained are cocrystals instead of salts? See the following references:

1) https://doi.org/10.1021/mp0601345 (Page 335-326)

2) https://doi.org/10.1039/C2CE26055G (Page 6363-6364)

3) https://doi.org/10.1021/acs.cgd.0c00210 (Page 5055-5058)

Response:

The ΔpKa rule (ΔpKa = pKa(base) ‒ pKa(acid)) is widely recognized in the literature for predicting whether an API and a guest molecule form a salt or cocrystal (Cruz-Cabeza et al. (2022). Faraday Discussions, 235, 446-466). When the ΔpKa value exceeds 4, the components tend to form a salt. A cocrystal is likely to occur if ΔpKa ≤ -1. However, due to the presence of the carboxamide moiety, carbamazepine is considered a non-ionizable substance (Alhalaweh, A. et al. (2012). Molecular pharmaceutics, 9(9), 2605-2612) that forms solely cocrystals. Several studies reported salt formation of carbamazepine through protonation of the carbonyl O atom of the carboxamide functional group that can be achieved under specific conditions and by using strong mineral (HCl, HBr) or sulfonic acids (methanesulfonic, benzenesulfonic acids, etc.) (Perumalla, S. R., & Sun, C. C. (2012). Chemistry–A European Journal, 18(21), 6462-6464; Eberlin, A. R., Eddleston, M. D., & Frampton, C. S. (2013). Acta Crystallographica Section C, 69(11), 1260-1266; Buist, A. R. et al. (2013) Crystal growth & design, 13(11), 5121-5127; Buist, A. R. et al. (2015) Crystal Growth & Design, 15(12), 5955-5962). However, the pKa value for the protonated carbamazepine species has not been determined and remains obscure. Thus, the ΔpKa rule cannot currently be applied to multicomponent crystals containing carbamazepine and an acidic coformer. We also have to emphasize that multiple examples of CBZ multicomponent crystals with various organic acids indicate that the drug only forms cocrystals, whereas the salt forms of CBZ are highly uncommon and limited to the cases mentioned above.

Round 2

Reviewer 3 Report

The study successfully combined carbamazepine (CBZ) with positional isomers of acetamidobenzoic acid, determined the structural and energetic features of resulting cocrystals, and evaluated three virtual screening methods for predicting cocrystallization outcomes. The hydrogen bond propensity model performed poorly, while molecular electrostatic potential maps and CCGNet showed comparable results. The cocrystallization reactions were found to be enthalpy-driven, with differences in dissolution behavior attributed to variations in thermodynamic stability.

1) The study underwent a revision process in which changes were made to improve its scientific rigor, and as a result of those revisions, the study is now considered to be scientifically sound.

2) The tables and figures presented in this study are readily comprehensible due to their clarity and accessibility.

3) The references utilized in this article are judiciously selected, as they include both contemporary and archival literature that provides corroborative evidence in support of the author's assertions.

4) I anticipate with interest the forthcoming investigations of the dissolution characteristics of CBZ cocrystals in your future research endeavors.